# Eccentric Compression Behaviors of Self-Compacting Concrete-Filled Thin-Walled Steel Tube Columns

**DOI:** 10.3390/ma16186330

**Published:** 2023-09-21

**Authors:** Yunyang Wang, Shengwei Sun, Liqing Zhang, Yandong Jia, Guang Yang, Meng Li, Wei Tan, Jianmin Qu

**Affiliations:** 1School of Civil and Architecture Engineering, Hunan University of Arts and Science, Changde 415000, China; 2China Construction Second Engineering Bureau Co., Ltd., Beijing 100054, China; ssw5256081@163.com; 3School of Civil Engineering and Architecture, East China Jiaotong University, Nanchang 330013, China; 4School of Civil and Architecture Engineering, Liaoning University of Technology, Jinzhou 121000, China; 5Hunan Bolian Engineering Testing Co., Ltd., Changde 415000, China

**Keywords:** eccentric compression, thin-walled steel tubes, failure characteristics, self-compacting concrete, mechanical performances

## Abstract

For the sake of solving sustainability issues and analyzing the complicated service force states, eccentric compression experiments on self-compacting concrete-filled thin-walled medium-length steel tube columns with a circular cross-section were carried out in the present study. Thereafter, the influence of the eccentric ratios and the wall thickness factors on the mechanical behavior and failure characteristics of both the eccentrically loaded and axially loaded columns was comprehensively analyzed. Finally, prediction formulas for the ultimate load of the columns under eccentric compression were proposed, and a comprehensive comparison of the ultimate loads between the predicted values and experimental values was also conducted. The results indicated that the typical failure characteristics of the eccentrically loaded columns presented lateral deflection together with buckling, while the axially compressed columns displayed expansion and rupture at local positions. Moreover, the ultimate loads of the eccentrically loaded columns decreased by 43.0% and 34.5% in comparison to the columns under axial compression, with the wall thickness factor decreasing from 116.7 to 46.7, respectively. Meanwhile, the ratios of the ultimate loads calculated using design codes to the tested values were in the range of 0.70~0.90, which demonstrated that the design codes could predict the ultimate loads conservatively. Additionally, the ratios of the ultimate loads calculated using the proposed formulas to the tested values were within the range of 0.99~1.08, implying that the proposed formulas were more accurate than the design codes. At the same time, the initial stiffness of the columns under eccentric compression was correspondingly lower than that of the columns undergoing axial compression. The lateral deflections along the height of the columns were almost symmetrical at different loading levels. This study could provide a meaningful approach for designing columns and facilitate their application in civil industry.

## 1. Introduction

Concrete-filled steel tubes have been extensively researched and applied in civil infrastructure, for example, long-span bridges, ultra-high-rise buildings, urban viaducts, and military facilities, due to their high strength, outstanding ductility, excellent seismic performance, and convenient construction [1,2]. Steel tubes can function as formworks for concreting, speeding up construction, and reducing the labor force, which may be a good way to solve sustainability issues during the construction process. Moreover, the performance of the concrete is enhanced by the restraining effect of the steel tube, while the concrete can also prevent the buckling of steel tubes [2,3,4,5].

Owing to their excellent advantages, research on concrete-filled steel tubes has been conducted worldwide [6,7,8,9,10,11]. Wei et al. demonstrated that the ultimate load of concrete-filled steel tube columns was increased upon increasing the strength of the concrete and steel tubes, while it was reduced upon increasing the ratio of the diameter to the thickness [12]. A model of lateral–axial strain in concrete was proposed by Zhu et al. to predict the interaction effect, and it could accurately predict the mechanical behavior of concrete-filled steel tubes [13]. Cao et al. demonstrated that the failure of columns under both monotonic and cyclic compressive loads was characterized by local buckling [4]. The influence of concrete strength and the D/t ratio on columns undergoing axial compressive loading was experimentally studied by Farid et al. The ultimate loads were also compared between the test results and calculation values [5].

The parameters of concrete-filled steel tubes undergoing eccentric loading examined in previous research including the strength of the materials, the ratio of the diameter to the thickness, the eccentricity ratio, the slenderness, and the cross-section shapes [14,15,16,17,18,19,20,21,22]. Slender columns with a circular cross-section containing concrete with a high strength undergoing eccentric loading were studied by Portolés et al.; the columns displayed good ductility, while the ultimate load was not improved [15]. Nevertheless, the ductility of the columns with a circular cross-section containing high-strength concrete under an eccentric load was reduced. Meanwhile, the ductility was improved in columns with a small diameter-to-thickness ratio [14]. The mechanical behavior and failure characteristics of long columns made from high-strength materials with a square cross-section undergoing eccentric loading have also been explored by experiments and numerical methods. Equations for predicting the ultimate load were proposed. The yield load was enhanced by high-strength steel, and the columns displayed failure characteristics of local buckling [17,19,20]. High-strength concrete-filled steel tube columns with a rectangular cross-section under eccentric loading displayed favorable ductility. The predicted ultimate loads using EC4 codes were 4% higher than the experimental values [18]. The ultimate loads of recycled-concrete-filled steel tube columns undergoing eccentric loading decreased upon increasing the substitution rate of the recycled aggregate [21]. The structural behavior of steel tube columns filled with ultra-high-performance concrete possessing both circular and square cross-sections under eccentric loading was studied, and the feasibility of predicting the ultimate load using design codes was also examined. The ultimate load decreased upon increasing the eccentricity and slenderness. The failure characteristics of the columns indicated in-plane bending [22].

In the past few years, concrete-filled thin-walled steel tube columns began to be researched and applied, following the fast development of high-strength steel tubes. The confinement on concrete is improved by high-strength steel tubes [7,23,24,25]. Thin-walled steel tubes filled with concrete are defined by a confinement coefficient less than 0.5 and a steel tube wall thickness less than 3 mm [26,27]. The efficiency of steel utilization can be increased, leading to a reduction in steel consumption and cost savings [28,29,30].

Li et al. studied the relationships between displacement and load and the failure modes of axially loaded columns using both experimental and finite element methods [24]. Liu et al. described the failure characteristics, relationships between load and displacement, and bond strength of axially loaded columns [31]. The effect of the D/t ratio and compression ratio, as well as the concrete strength, on the seismic performance of columns under cyclic lateral loading together with compressive loading was studied by Wang et al. The steel tubes were severely ruptured, and the infill concrete was almost crushed, representing the main failure modes. Additionally, the ductility, stiffness degradation, skeleton curves, and capacity of energy dissipation were also discussed [32]. Jiang et al. obtained the relationships between load and displacement, failure modes, and ultimate load of the components of concrete-filled thin-walled steel tubes under a bending load and proposed a model to predict the bending strength [33]. The failure characteristics and interaction effect of the columns under a compressive load were obtained by Wang et al. using the finite element method [34]. The prediction of the ultimate loads of rectangular columns undergoing axial loading was conducted by Cakiroglu et al. using machine learning models. The predicted ultimate loads presented a high accuracy of up to 98.3% [35].

Self-compacting concrete was first developed in the 1980s, and it can be compacted by its own weight and need not extra vibration. It also presents advantages such as outstanding workability, good mechanical performance, and durability [36,37,38,39]. Thus, the shortcomings such as lack of labor force and uneven consolidation of concrete can be overcome, and costs will be saved. This concrete maybe the best choice application in concrete-filled steel tubes [36,38,40,41].

Nevertheless, the primary research on the columns mostly concern the axially loaded behaviors. Wang et al. pointed out that the ACI code predicts the most conservative ultimate load of the axially loaded columns while the CECS code gives the most precise ultimate load of the same columns [42]. It was demonstrated that the local buckling of the axially loaded columns with X section can be improved by the intermediate stiffeners. Moreover, the predicting ultimate load by AISC code presented a conservative result [43]. However, little study has been conducted on the eccentrically loaded columns. In fact, the columns in practical engineering are more likely to undergo complicated force states, such as eccentric compressive load is the most common force style [44]. Therefore, self-compacting concrete-filled thin-walled steel tube (SCCTST) medium-length columns with circular cross-sections were first developed in the present study. And then, eccentric loading experiments of the columns were conducted. Thereafter, the effect of the eccentric ratios and the wall thickness factor on failure characteristics and mechanical behaviors of the columns undergoing both eccentric and axial compression were analyzed comprehensively. The aim of this paper is to provide meaningful approach for designing and facilitate the application of the SCCTST columns in civil infrastructures.

## 2. Experimental Methods

### 2.1. Raw Materials and Specimen Preparation

Four of the SCCTST column specimens with circular cross-section were manufactured, wherein two of the columns undergo eccentric compressive load, and the other two columns withstand axial compressive load. The key parameters are the eccentric ratios and the wall thickness factor β, which is defined as the ratio of D/t. Steel tubes with wall thickness of 1.2 mm and 3.0 mm were adopted. Two kinds of eccentric ratios included 0 and 0.29. Outer diameter and length of the columns were designed as 140 mm and 700 mm, respectively. Table 1 shows the numbers and dimensions of the columns. Therein, D represents the outer diameter, t stands for the wall thickness of steel tubes, and L represents length of the columns. Letters A and E in Table 1 refer to the axially loaded columns and eccentrically loaded columns, respectively. Self-compacting concrete is described as N, and its mix proportion is shown in Table 2. T1 and T2 stand for the wall thickness of the steel tube being 1.2 mm and 3.0 mm, respectively. In addition, A_s_ and A_c_ represent the area of steel tube and concrete, respectively, and e and r represent the load eccentric distance and outer radius of the columns, respectively. The values of the wall thickness factor of 116.7 and 46.7 correspond to wall thickness of 1.2 mm and 3.0 mm, respectively.

Two square endplates were welded on the columns’ ends. The thickness and side length of the square endplates were 20 mm and 180 mm, respectively. A square steel endplate was welded onto the bottom end of the columns before casting concrete, and it could function as a mold together with the steel tubes during the concrete casting process. The geometric center of the endplate and the steel tubes must remain in a vertical line to ensure the load can apply to the columns evenly. External vibration was not used during the whole casting process. The columns were cured in a laboratory with relative humidity and temperature of 95% and 20 °C, respectively, lasting for 28 d after casting the concrete. However, shrinkage about 2 mm was generated on the infill concrete after curing for 28 d. Therefore, high-strength cement paste was used to fill the uneven space generated by the concrete shrinkage. The top end was welded to another steel endplate before testing. Preparation for the SCCTST columns is shown in Figure 1.

### 2.2. Material Properties

The mechanical performances of the steel tubes were tested by experiments which are listed in Table 3. The testing method is in accordance with the standard [45]. Three strips were cut from each type of the steel tubes used for testing the mechanical performances.

Both the concrete used to fabricate specimens to test the mechanical performances and the concrete used to fill in the steel tubes were mixed together. The concrete specimens used to test mechanical performances contained three cubic concrete specimens and six cuboid concrete specimens. The side length of the cubic specimens is 150 mm. Additionally, the dimensions of the cuboid are 150 mm × 150 mm × 300 mm. The concrete specimens were cured in the laboratory with similar conditions to the SCCTST columns, and plastic film covers were placed on top of the columns after concreting. The mechanical performances of concrete were tested according to the standard [46]. Its modulus of elasticity and compressive strength are 25.7 GPa and 54.4 MPa, respectively.

### 2.3. Test Setup and Procedure

The SCCTST columns under eccentric and axial compressive load after curing for 28 d were conducted by a universal testing machine with maximum load up to 3000 kN. The eccentric load was first exerted on the spherical hinges, as shown in Figure 2. And then, it was transferred to the load-bearing plates. Finally, the load was transferred from the bearing plates to the endplates of the columns. In addition, the axial load was directly exerted on the endplates. The spherical hinge was made from a steel cylinder; its diameter was 100 mm and length was 180 mm. A cuboid steel plate with side length and thickness of 180 mm and 30 mm, respectively, was used as the load-bearing plate. Figure 2 displays the locations and arrangements of the loading devices. The strains in the transverse and axial directions at the middle position of the columns were numbered from 1 to 8 and they were measured by strain gauges. Among them, strain gauges 1 to 4 and 5 to 8 corresponded to the strains in the transverse and axial directions, respectively, as shown in Figure 2c. The lateral deflections and vertical displacements were tested by displacement meters corresponding to numbers of 1 to 3 and 4 to 5, respectively, as shown in Figure 2d,e. Here, two of the displacement meters were placed in the bottom end to measure the displacements in the axial direction. The other three displacement meters were used to measure the lateral deflections and were symmetrically placed in the left at the middle height of the eccentric loaded columns to measure the lateral deflections.

Preliminary loading test was carried out before formal loading to ensure the load could compact on the endplates of the columns evenly. The loading amplitude in the preliminary loading test process was 15 kN, and the loading rate was 0.5 kN/min. Loads were applied to the columns continuously after the preliminary loading stage. During the formal testing process, one-fifteenth of the predicted ultimate load was set as the loading interval and the loading rate was 0.5 mm/min, then 3 min was maintained after each load interval had finished. The test was stopped after the load decreased to 70% of the ultimate load. The total loading time of one column was about 2 h. The data on loads, displacements, and strains during the loading process were automatically collected.

## 3. Results and Analysis

### 3.1. Characteristics at Failure Stage

Figure 3 displays the failure characteristics of the columns withstanding both eccentric and axial compressive load, and it demonstrated that the main failure characteristics of E–N–T1 and E–N–T2 withstanding eccentric load are both lateral deflection and buckling. Correspondingly, no evident lateral deflections and buckling were found on the columns withstanding axial compressive load. Thus, both the axial compression columns A–N–T1 and A–N–T2 displayed primary failure modes as expansion and rupture on the bottom end and middle height of the columns. With the wall thickness factor β decreasing from 116.7 to 46.7, the numbers of expansion were decreased and the position was moved from the bottom end to the middle height. Nevertheless, the bulking and rupture of the columns withstanding eccentric compressive load was less evident than that of the columns withstanding axial compressive load. Additionally, the lateral deflections and buckling of the columns undergoing eccentric load moved from the bottom end to the middle height with increase of the wall thickness. A similar failure characteristics were also observed by Lee. However, the local expansion appeared at the upper end when the columns underwent axial load, while it was generated at the middle height when the columns were subjected to eccentric load [16].

No evident changes were present on the surface of the columns before loading to 80% of the ultimate load. And then, a small amount of bulking and buckling was generated on the surface of the columns corresponding to the columns withstanding eccentric compressive load. The bulking and buckling was more evident when loading to the ultimate load.

### 3.2. Relations between Displacement and Load

The relations between displacement and load in the axial direction of the columns and the ultimate loads from experiments ($N_{ue}$) are displayed in Figure 4. It demonstrated clearly that with increasing wall thickness, the ultimate loads withstanding both axial and eccentric load were increased. The ultimate loads of the columns A–N–T1, E–N–T1, A–N–T2, and E–N–T2 were 1331.0 kN, 759.0 kN, 1627.8 kN, and 1065.8 kN, respectively. Compared with the columns A–N–T1 and A–N–T2 withstanding axial compression with the wall thickness factor decreasing from 116.7 to 46.7, the ultimate loads of E–N–T1 and E–N–T2 withstanding eccentric compressive load were correspondingly decreased by 43.0% and 34.5%, respectively. In addition, the ultimate loads corresponding to the axially and eccentrically loaded columns increased by 22.3% and 40.4%, respectively, with decrease of the wall thickness factor β from 116.7 to 46.7.

Theoretical calculation formulas on the ultimate loads of the columns subjected to eccentric compression were set up from fitting of the experimental data of load and displacement which are displayed in Figure 4a. As shown in Figure 4a, the fitting curves coincided well with the tested curves. R^2^ values of the fitting curves related to wall thickness factors 116.7 and 46.7 were 0.9991 and 0.9988, respectively. This implied that the proposed formulas can well predict the relations of load and displacement. Equations (1) and (2) are the proposed calculation formulas of the ultimate loads corresponding to different wall thickness factors 116.7 and 46.7, respectively.
(1)$$F=-2.85\delta +83.29\delta^{2}-13.49\delta^{3}+0.54\delta^{4}$$
(2)$$F=-104.07\delta +139.01\delta^{2}-19.70\delta^{3}+0.76\delta^{4}$$
where F stands for the ultimate load and δ represents the displacement.

The ultimate load of the columns was obtained using calculation formulas of the design codes CECS 28:90 [47] and DL/T 5085-1999 [48]. Table 4 lists the comparison results between the tested results and calculated values using design codes and proposed prediction formulas. As shown in Table 4, the calculated ultimate loads using the design codes are correspondingly lower than those of the tested values. The ratios of the ultimate loads calculated by design codes of CECS and DL/T to the tested values are under the ranges of 0.79~0.90 and 0.70~0.83, respectively. This implied that the design codes can conservatively predict the ultimate loads. Meanwhile, the ultimate loads predicted using the proposed formulas corresponding to the wall thickness factor of 116.7 and 46.7 are 751.6 kN and 1148.7 kN, respectively. The corresponding ratios of the ultimate loads calculated using the proposed formulas to the tested values are 0.99 and 1.08, respectively. This demonstrated that the proposed formulas predicting the ultimate loads of the SCCTST columns are more accurate than those of the design codes.

Table 5 summarizes the performances of the concrete-filled steel tube columns in previous research. As shown in Table 5, the slenderness is within the range of 2.1~10.7, and most of the slenderness is under the range of 2.1~3.6. Meanwhile, the wall thickness is within the range of 0.92 mm~5.6 mm. Moreover, the ultimate loads are under the scope of 391 kN~3031 kN. Therefore, the ultimate loads are generally decreased with increase of the slenderness. Additionally, the ultimate loads are influenced by wall thickness, diameter, length, slenderness, strength of concrete and steel tube, and eccentricity.

The descent phase of the curves on load–displacement of the axial compression columns is not evident, while the descent phase of the columns withstanding eccentric compressive load is clear and more gentlely. Meanwhile, the descent phase is more gently with thicker wall thickness. The initial stiffness $K_{n}$ is expressed as Equation (3) [44].
(3)$$K_{n}=\frac{0.4N_{ue}}{\Delta }$$
where $N_{ue}$ and Δ represent the ultimate load from experimental results and the displacement at 0.4$N_{ue}$, respectively. $D_{ue}$ is the displacement at the ultimate load. Figure 4c,d display the comparison of the initial stiffness $K_{n}$ and $D_{ue}$ between the columns withstand eccentric load and axial load, respectively. As shown in Figure 4c, in the comparison of the initial stiffness of the columns A–N–T1 and A–N–T2, the initial stiffness of E–N–T1 and E–N–T2 is correspondingly reduced by 14.1% and 25.4%, respectively. Moreover, the initial stiffness corresponding to axial load and eccentric load is increased by 37.3% and 19.3%, respectively, as the wall thickness factor decreases from 116.7 to 46.7.

Figure 4d presents the displacements at the ultimate loads of A–N–T1, E–N–T1, A–N–T2, and E–N–T2 which are 6.67 mm, 5.46 mm, 6.42 mm, and 6.67 mm, respectively. The displacements of the eccentrically loaded columns are decreased by 18.1% and increased by 3.9%, respectively, corresponding to the wall thickness factors 116.7 and 46.7, compared with the displacements of the axially loaded columns. In brief, the initial stiffness and the displacement at the ultimate load are approximately growing with increasing of the wall thickness, while they are decreased when loading types changes from axial to eccentric. The displacements at the ultimate load are within the range of 7.14 mm~18.5 mm [16], which is higher than that of the displacements at the ultimate loads in this paper. This is because their wall thicknesses are 6.0 mm and 12.0 mm higher than that of this paper.

### 3.3. Relations between Load and Strain

Figure 5 presents the relations between load and strain of SCCTST columns undergoing eccentric and axial load. The yield strain $\varepsilon_{y}$ of the steel tubes is expressed by Equation (4),
(4)$$\varepsilon_{y}=\frac{f_{y}}{E_{s}}$$
where $f_{y}$ and $E_{s}$ stand for the steel tube properties of yield strength and modulus of elasticity, respectively. The axial and transverse strains are described as negative and positive, respectively. As shown in Figure 5, the strains develop nonlinearly with increase of the load. However, the strains in the axial direction are developed more quickly than the strains in the transverse direction. The strain ε_ue_ corresponding to the ultimate load of the eccentric loading columns is larger than that of the axial loading columns. Figure 6 shows the comparison on the yield strains and ε_ue_. As shown in Figure 6, ε_y1_ and ε_y2_ stand for the yield strains of steel tubes corresponding to the wall thickness factor of 116.7 and 46.7, respectively, which are 1906 με and 1774 με, respectively.

The average ε_ue_ in the axial and transverse directions is correspondingly larger and smaller than that of the yield strain. This indicated that the confinement in the transverse direction is weak under the ultimate load. The average ε_ue_ of the column A–N–T1 at the axial and the transverse directions is −2240 με and 1086 με, respectively. Compared with ε_y1_, the average ε_ue_ of A–N–T1 at axial and transverse directions is correspondingly increased by 17.5% and decreased by 43.0%, respectively. Additionally, the average ε_ue_ of the column E–N–T1 at the axial and the transverse directions is −2359 με and 1022 με, respectively. Similarly, compared with the yield strain ε_y1_, the average ε_ue_ of E–N–T1 at axial and transverse directions is correspondingly increased by 23.8% and decreased by 46.4%, respectively. The average ε_ue_ of the column A–N–T2 at the axial and the transverse directions is −2316 με and 1089 με, respectively. Compared with the yield strain ε_y2_, the average ε_ue_ of A–N–T2 at axial and transverse directions is correspondingly increased by 30.6% and decreased by 27.3%, respectively. The average ε_ue_ of the column E–N–T2 at the axial and the transverse directions is −2958 με and 1263 με, respectively. Compared with the yield strain ε_y2_, the average ε_ue_ of E–N–T2 at axial and transverse directions is correspondingly increased by 66.7% and decreased by 28.8%, respectively.

The ε_ue_ of the axially loaded columns is approximately increased upon increasing the wall thickness. Compared with A–N–T1, the ε_ue_ at axial and transverse directions of A–N–T2 is increased by 3.4% and 18.7%, respectively. Compared with E–N–T1, the ε_ue_ at axial and transverse directions of E–N–T2 is increased by 25.4% and 23.6%, respectively.

The ε_ue_ in the axial and transverse directions of the columns withstanding eccentric load is correspondingly larger and smaller than that of axially loaded columns. Compared with A–N–T1, the ε_ue_ at axial and transverse directions of E–N–T1 is correspondingly increased by 5.3% and decreased by 5.9%, respectively. Compared with A–N–T2, the ε_ue_ at axial and transverse directions of E–N–T2 is correspondingly increased by 27.7% and decreased by 2.0%, respectively. 

### 3.4. Relations between Load and Lateral Deflection

Figure 7 displays the relations between load and lateral deflection of the SCCTST columns subject to eccentric load. Figure 7 describes that the lateral deflection of E–N–T1 and E–N–T2 almost increases linearly when the load increases at 70% of the ultimate load. And then, the growth rate of the lateral deflections is more quickly than that of the load, and the lateral deflection is increased nonlinearly with the load. The lateral deflection is gradually decreased with continued loading after the ultimate load. As the wall thickness factor decreases from 116.7 to 46.7, the load–lateral deflection curves of the descending branch are more gentlely. Av1 and Av2 represent the average lateral deflections withstanding the maximum load of E–N–T1 and E–N–T2, respectively. The average lateral deflection f_ue_ withstanding the maximum load of E–N–T1 and E–N–T2 is 3.8 mm and 10.0 mm, respectively. The average lateral deflection f_ue_ of E–N–T2 is increased by 162.3% compared with that of E–N–T1.

Figure 8 describes the relations between load and lateral deflection with different relative height of the columns subject to eccentric load. It can be seen from Figure 8 that the lateral deflection along the column height is almost symmetrical at different loading levels. Moreover, the symmetry of the lateral deflection of E–N–T2 at different loading levels is more evident compared with that of E–N–T1, as shown in Figure 8b. It is indicated that the symmetry of the lateral deflection along the column height is more evident with increase of the wall thickness. This phenomenon indicates that the confinement and the force state will be improved with thicker wall thickness.

### 3.5. Coefficient of Displacement Ductility

Equation (5) expresses the displacement ductility coefficient μ of the SCCTST columns [57].
(5)$$\mu =\frac{\Delta_{u}}{\Delta_{y}}$$
where $\Delta_{y}$ stands for the yield displacement at 60% of the ultimate load, and $\Delta_{u}$ stands for the displacement of the load diminished to 85% of the ultimate load. The displacement ductility coefficients of the columns are displayed in Table 6, and their comparison is shown in Figure 9. As shown in Table 6, the coefficients of displacement ductility of A–N–T1, E–N–T1, A–N–T2, and E–N–T2 are 1.57, 2.00, 1.69, and 2.35, respectively. The displacement ductility coefficients in this paper are comparable with those of previous researches, which are within the range of 1.42~3.5 [28]. This indicates that the SCCTST columns possess favorable ductility. For a good composition between steel tube and concrete, the properties of the concrete will be improved significantly. Moreover, buckling inward of the steel tube can also be prevented by the concrete [2,3,4,5].

Compared with A–N–T1, the coefficient of displacement ductility of E–N–T1 is increased by 27.1%, as shown in Figure 9. Similarly, the coefficient of displacement ductility of E–N–T2 is increased by 39.2% compared with A–N–T2. It indicates that the SCCTST columns under eccentric compression present better ductility compared with the columns under axial compression. Moreover, the coefficient of displacement ductility of A–N–T2 is 7.2% higher than that of A–N–T1, and the coefficient of displacement ductility of E–N–T2 is 17.4% higher than that of E–N–T1. This also indicated that the SCCTST columns with thicker wall thickness present better ductility.

## 4. Conclusions

In order to reduce energy consumption and analyze the complicated service states, SCCTST columns were first developed. And then, their failure modes and mechanical performances withstanding axial and eccentric compressive load were investigated. Thereafter, the influence of eccentric ratios and wall thickness factor on failure characteristics, relations between load and displacement, relations between load and strain, initial stiffness, displacements/strains corresponding to the ultimate loads, relations between load and lateral deflection, and coefficients of displacement ductility of the SCCTST columns were analyzed in depth. Finally, prediction formulas of the ultimate loads were proposed, and the calculated ultimate loads were compared with the tested values. Conclusions can be drawn as follows.

Both lateral deflection and buckling were the main failure characteristics of the eccentrically loaded columns, while the columns withstanding axial load displayed bulking and rupture of the columns. The amount of bulking decreased upon increasing wall thickness.The descent phase of the relations between load and displacement of the eccentrically loaded columns was more gently compared with the axially loaded columns. The descent phase became more gently with increase of the wall thickness. In comparison with the axially loaded columns, the ultimate load of the eccentrically loaded columns decreased by 43.0% and 34.5%, respectively, corresponding to the wall thickness factors 116.7 and 46.7.Theoretical calculation formulas on predicting the ultimate loads of the columns subjected to eccentric compression were proposed, and the proposed curves coincided well with the tested curves. Meanwhile, the ratios of the ultimate loads calculated using design codes to the tested values were under the range of 0.70~0.90. This implies that the design codes can conservatively predict the ultimate loads. Additionally, the ratios of the ultimate loads calculated using the proposed formulas to the tested values were within the range of 0.99~1.08; it was demonstrated that the proposed formulas predicting the ultimate loads of the SCCTST columns withstanding eccentric compression are more accurate than those of the design codes.Initial stiffness of eccentrically loaded columns was reduced by 14.1% and 25.4%, respectively, compared with the axially loaded columns corresponding to wall thickness factors 116.7 and 46.7. When decreasing the wall thickness factor from 116.7 to 46.7, initial stiffness of the eccentrically loaded columns increased by 19.3%, while the displacements at the ultimate loads of the eccentrically loaded columns decreased by 18.1% and increased by 3.9%, respectively.The strains developed nonlinearly with increases of the load. Compared with the yield strain, the average strain ε_ue_ of the columns withstanding eccentric load in axial and transverse directions was correspondingly increased by 23.8%/66.7% and decreased by 46.4%/28.8%, respectively. The ε_ue_ was almost increased with increasing wall thickness.Lateral deflections of the columns withstanding eccentric load proximately increased linearly with load increasing up to 70% of the maximum load. After that, the increase rate of load was slower than that of the lateral deflection, and the relations between lateral deflection and load were nonlinear. The average lateral deflection corresponding to the ultimate load with wall thickness of 3.0 mm increased by 162.3% compared with that with the wall thickness of 1.2 mm. The lateral deflection along the columns’ height was proximately symmetrical at different loading levels. The symmetry of the lateral deflection and the ductility was more evident with increasing wall thickness.

Quantity investigations on concrete-filled steel tubes have been conducted by experimental and numerical analyses. However, little studies have been carried out on the SCCTST columns subjected to eccentric compression because the mechanical behaviors of the SCCTST columns under these situations are considerably complex when applied to practical engineering. Thus, guidance for designing and optimizing the SCCTST columns withstanding eccentric compressive load for application in civil infrastructures should be founded in the future research.

## Figures and Tables

**Figure 1 materials-16-06330-f001:**
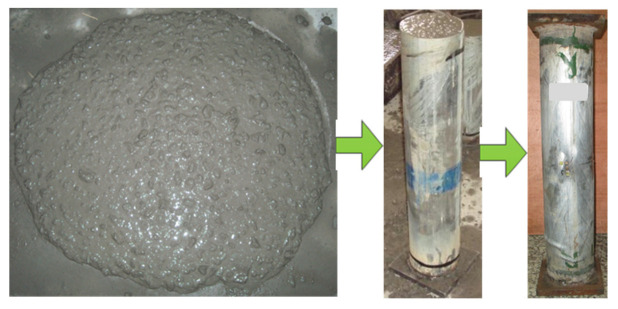
Preparation for the SCCTST columns.

**Figure 2 materials-16-06330-f002:**
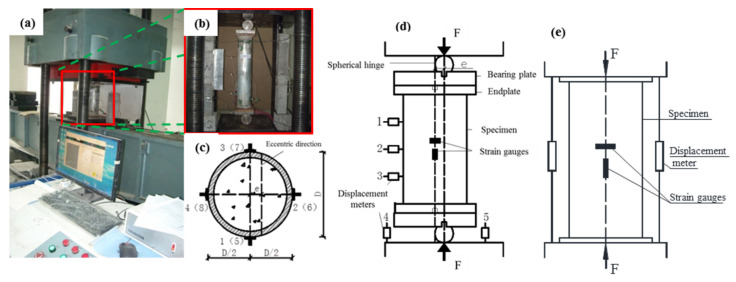
Schematic of loading devices and loading process of the SCCTST columns. (**a**) The test machine. (**b**) The photograph of the eccentrically loaded columns. (**c**) The locations and arrangements of the strain gauges in cross-section area. (**d**) The schematic of the columns withstand eccentric load. (**e**) The schematic of the columns withstand the axial load.

**Figure 3 materials-16-06330-f003:**
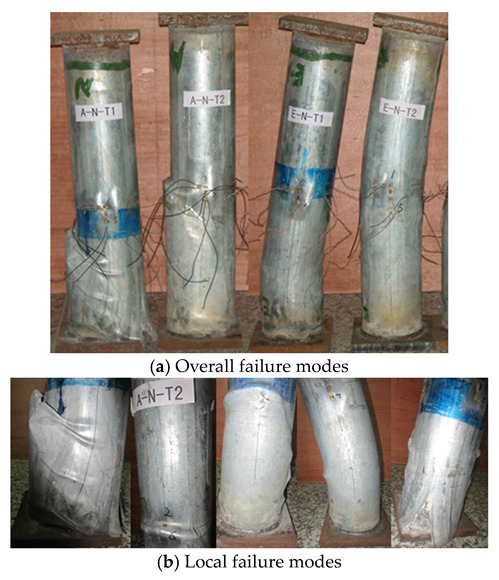
Failure modes of the columns.

**Figure 4 materials-16-06330-f004:**
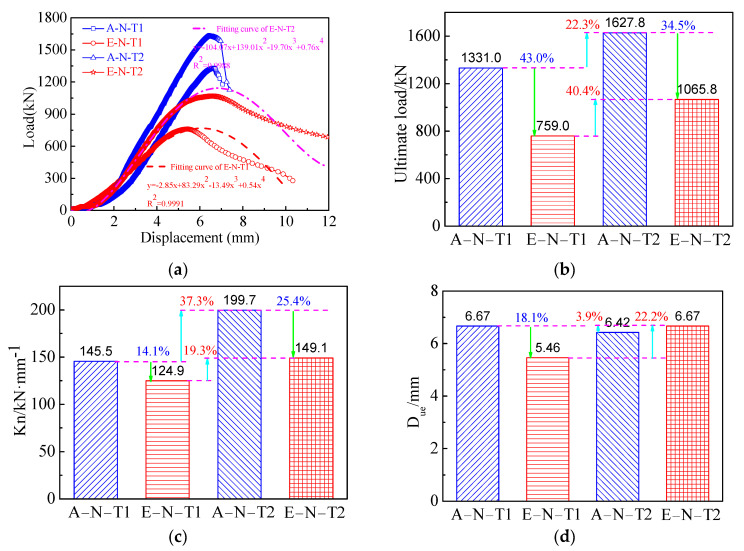
Relations of displacement and load of the SCCTST columns. (**a**) The tested and fitting curves on load–displacement. (**b**) The ultimate loads. (**c**) Initial stiffness. (**d**) Displacement corresponding to the ultimate load.

**Figure 5 materials-16-06330-f005:**
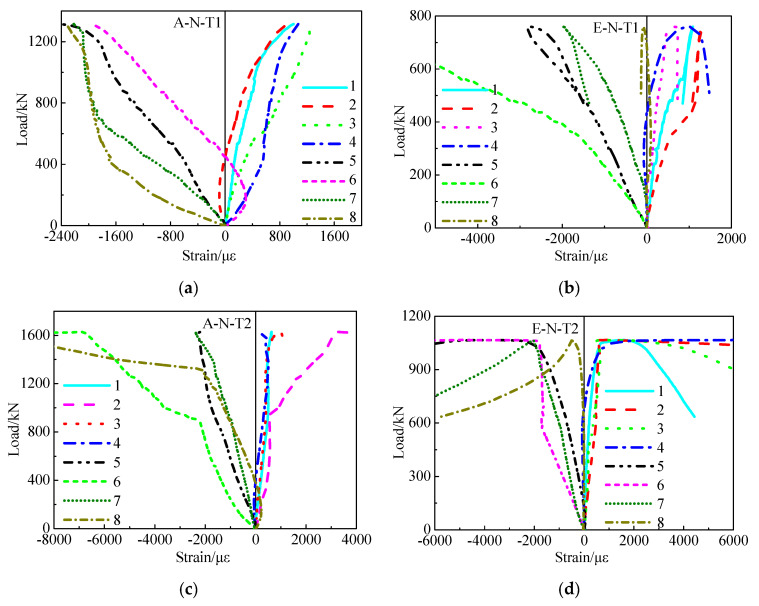
Relations between load and strain. (**a**) Relations between load and strain of A–N–T1. (**b**) Relations between load and strain of E–N–T1. (**c**) Relations between load and strain of A–N–T2. (**d**) Relations between load and strain of E–N–T2.

**Figure 6 materials-16-06330-f006:**
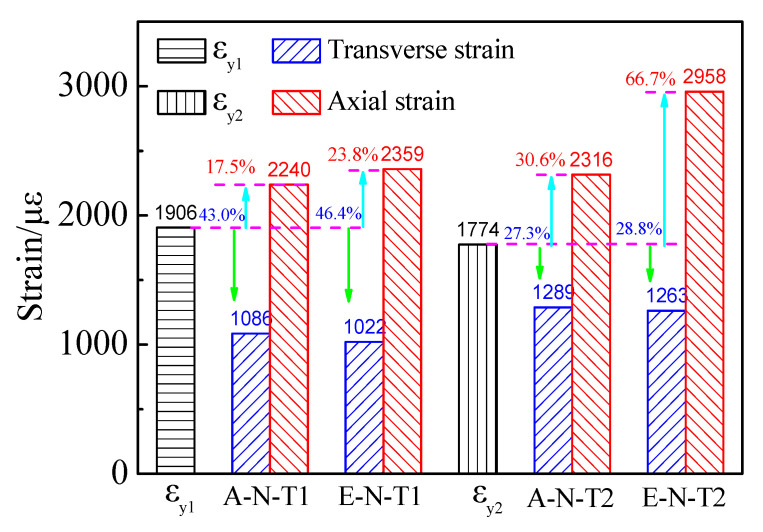
Comparison on the yield strains and the strains corresponding to the ultimate loads.

**Figure 7 materials-16-06330-f007:**
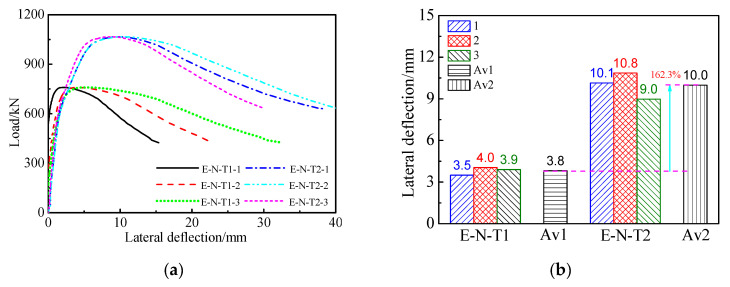
Relations between load and lateral deflections. (**a**) Load–lateral deflection curves. (**b**) Lateral deflection corresponding to ultimate load.

**Figure 8 materials-16-06330-f008:**
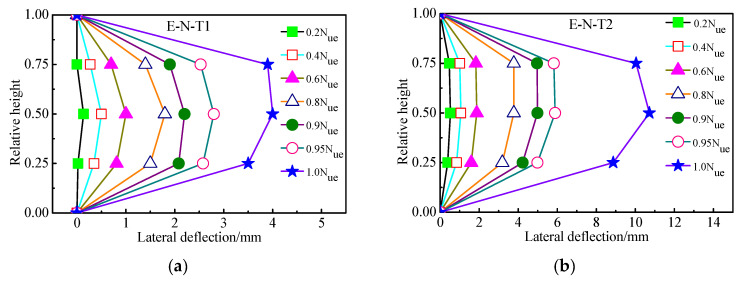
Relations between load and lateral deflection with different relative height of the columns subject to eccentric load. (**a**) E–N–T1. (**b**) E–N–T2.

**Figure 9 materials-16-06330-f009:**
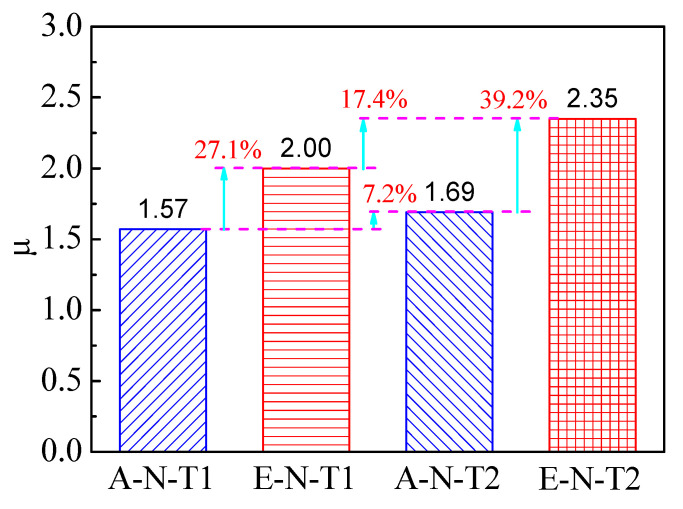
Comparison of the coefficients of displacement ductility.

**Table 1 materials-16-06330-t001:** Dimensions of the SCCTST columns.

Numbers	D × t × L (mm^3^)	L/D	A_s_ (mm^2^)	e (mm)	e/r	A_s_/A_c_ (%)
A–N–T1	140 × 1.2 × 700	5.0	519.2	0	0	3.4
E–N–T1	140 × 1.2 × 700	5.0	519.2	20	0.29	3.4
A–N–T2	140 × 3.0 × 700	5.0	1290.5	0	0	8.4
E–N–T2	140 × 3.0 × 700	5.0	1290.5	20	0.29	8.4

**Table 2 materials-16-06330-t002:** The mix proportion of the concrete (unit: kg/m^3^).

Coarse Aggregate	Fine Aggregate	Cement	Water	Fly Ash	Superplasticizer (wt.%)
769	837	402	199	122	0.4

**Table 3 materials-16-06330-t003:** Mechanical performances of the steel tubes.

Modulus of Elasticity (GPa)	Yield Strength (MPa)	Poisson’s Ratio	Wall Thickness (mm)
181.0	345.0	0.30	1.2
202.0	358.3	0.28	3.0

**Table 4 materials-16-06330-t004:** Comparison on tested and calculated ultimate loads.

Numbers	*N_ue_* *	CECS		DL/T		Proposed Formulas	
	(kN)	*N_uc_* * (kN)	*N_uc_*/*N_ue_*	*N_uc_* * (kN)	*N_uc_*/*N_ue_*	*N_uc_* * (kN)	*N_uc_*/*N_ue_*
A–N–T1	1331.0	1047.8	0.79	1059.9	0.80	-	-
E–N–T1	759.0	679.6	0.90	555.3	0.73	751.6	0.99
A–N–T2	1627.8	1426.0	0.88	1356.7	0.83	-	-
E–N–T2	1065.8	918.7	0.86	713.1	0.70	1148.7	1.08

*N_ue_* *: tested ultimate load. *N_uc_* *: calculated ultimate load.

**Table 5 materials-16-06330-t005:** Performances of concrete-filled steel tube columns.

D (mm)	t (mm)	L (mm)	Slenderness	*f_c_* * (MPa)	*f_y_* * (MPa)	*e* * (mm)	*N_ue_* * (kN)	References
114	1.7–2.09	394–402	3.5	29.2–35.6	300.3	0	557–688	[49]
133–140	2.64–4.66	400–420	3.0	36.9–52.9	302–335.3	0	1070–1749	[50]
138–170.6	2.79–2.86	420–510	3.0	36.3–40.0	339.6–388.5	0	1147.5–1607.4	[51]
139	0.92–1.92	500	3.6	32.6	238	0	505.6–931.9	[52]
114–167	3.1–5.6	250–350	2.1–2.2	44–60	300	0	1042–1873	[5]
114–115	3.84–5.02	298–300	2.6	31–104.9	343–365	0	929–1787	[53]
100–200	3.0	300–600	3.0	50	303.5	0	708–2594	[54]
133–159	3.1–6.2	399–477	3.0	75.1–80.7	331.7–392	0	2185–3031	[55]
150–180	3.0	450–540	3.0	59.3	324.4	0–30	689–1618	[44]
140	3.63	500–1500	3.6–10.7	30–60	233	0–60	391–773	[56]

Note. *f_c_* *: compressive strength of concrete. *f_y_* *: yield strength of steel tube. *e* *: eccentricity. *N_ue_* *: tested ultimate load.

**Table 6 materials-16-06330-t006:** Coefficients of displacement ductility.

Numbers	Δ*_y_* (mm)	Δ*_u_* (mm)	*µ*
A–N–T1	4.49	7.07	1.57
E–N–T1	3.20	6.40	2.00
A–N–T2	4.21	7.10	1.69
E–N–T2	3.60	8.44	2.35

## Data Availability

Not applicable.

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
