# Peer review of "Eccentric Compression Behaviors of Self-Compacting Concrete-Filled Thin-Walled Steel Tube Columns"

_materials, 2023, doi:10.3390/ma16186330_

Round 1
Reviewer 1 Report
The authors analyzed the behavior of self compacting concrete filled steel tubes under eccentric compressive loading. Full scale experiments have been carried out. The effect of wall thickness on the load carrying capacity has been investigated. The following points should be considered by the authors:
1) The authors should clearly indicate in the Abstract whether the paper is dealing with slender columns or stub columns.
2) The authors carried out an experimental program with concrete filled steel tubes. However this is not mentioned until after the Introduction section. The authors should mention in the Abstract and in the Introduction that full scale experiments have been carried out. Currently it is only mentioned that the behavior of SCCTST columns has been analyzed and the information about the type of analysis is missing from the Abstract and the Introduction.
3) The cross-section type of the SCCTST columns should be mentioned in the Abstract and in the Introduction. It should be made clear whether the experimental study deals with circular columns or rectangular columns.
4) The authors should mentioned the following related recent publications in the literature review:
https://doi.org/10.3390/su142114640
https://doi.org/10.1016/j.conbuildmat.2022.129227.
5) On page 4: "after curving 28d are conducted" → Should be curing 28d.
6) On page 5: "Thus, bulking in section and the steel tube even ruptured on bottom end and middle height is the mainly characteristics of the failure modes of the axial compressive loaded columns A-N-T1 and A-N-T2." → The meaning of this sentence is unclear. Particularly the usage of the term "bulking" is unusual in this context. Please revise.
7) On page 10: "This indicates that the SCCTST columns possess favourable ductility" → Here the authors should explain how they define "favorable" and why the columns can be deemed having favorable ductility.
8) The Conclusion section should emphasize the significance of this research project. The first sentence of Section 5. Conclusions states that "In order to reduce energy consumption and analyze the complicated service states, SCCTST columns are developed first." What the authors mean with this sentence is not very clear. The relationship of the current paper with energy consumption reduction and complicated service states should be clarified.
9) It is advisable to compare the experimental results with equations available in the literature. The authors should report whether there are equations suitable for self compacting concrete and for the geometry of the specimens in the literature or not.
English should be edited and revised.
Reviewer 2 Report
This paper provided the results of experimental research on the evaluation of the compression performance of a filled steel tube column subjected to an eccentric load. In order to improve the quality of the paper, the reviewer requests the following items to be revised
1) An analysis of a recent existing study on the compression performance of a CFT column subjected to an eccentric load should be added.
2) In particular, it is somewhat limited in suggesting conclusions based on the test results of four specimens. Therefore, it is necessary to draw conclusions by analyzing the results of existing studies.
Reviewer 3 Report
In their study, the Authors present the results of research on the strength of self-compacting columns made of thin-walled pipes filled with concrete. The manuscript has been written in an understandable way and the presented results are up-to-date. Nevertheless, there are some issues in the study that require clarification or supplementation.
1. The description of the purpose of the research, presented at the end of the first chapter, requires more detail and extension. It is true that the Authors present the aim of the research in a very general way and without referring to the results of research carried out in this area.
2. Wouldn't it be better to introduce a pipe wall thickness factor depending on its diameter? Such an approach would make it possible to generalize the research results.
3. The study presents test results for only one height of the samples. Due to the high slenderness of the columns, and thus the risk of buckling of such elements, it is crucial to determine the effect of the slenderness of the columns on their load capacity / strength.
4. In the conducted tests, a load was applied to the samples, which acted in the direction of the column axis. Were there any tests carried out in which the load would be applied at different angles to the axis of the column? In real conditions, it is very difficult to keep the front surfaces of the keels perpendicular to their axis, and thus the axial load on the columns.
5. How can the heterogeneity of the concrete filling the pipe affect the obtained results? Has research been conducted in this area?
6. Will a greater increase in pipe wall thickness affect the strength of the column? Similarly, do smaller wall thicknesses than those analyzed affect the strength of the columns?
7. At what thickness of the pipe wall strength properties do not change or change to a small extent?
8. Was a variant where the pipes would be filled with reinforced concrete considered? What effect can this have on the strength of such columns?
Round 2
Reviewer 1 Report
The manuscript has been sufficiently improved.
Reviewer 2 Report
Rather than a simple description of existing research results in the text of the paper, it is thought that the main focus of this study should be to present a method for calculating the axial load of a CFT column subjected to an eccentric load by combining the results and the experimental results of this study.
Sufficient supplementation is required for this.
Reviewer 3 Report
Thank you very much for the explanations and discussion. The explanations provided by the authors are sufficient.
